# SEMANTICDPP: EFFICIENT UNCERTAINTY QUANTIFICATION IN LLMS

## ABSTRACT

Accurately quantifying uncertainty is crucial for ensuring the reliability of model outputs and enabling informed downstream decision-making. However, the output space of large language models (LLMs) is so large that traditional methods break down in this setting; yet, LLMs have been shown to respond to prompts confidently with confabulations, fabrications, "hallucinations", and other erroneous information. While methods designed specifically for LLMs exist, there is a gap and need for approaches which accurately quantify uncertainty efficiently. In this work, we introduce SEMANTICDPP, an efficient method for quantifying the uncertainty corresponding to the semantic variation in outputs of a large language model based on internal activations. Building upon prior work, we show that this uncertainty is highly indicative of whether or not a model will answer correctly in question answering (QA) tasks—which provides a useful signal to determine whether to abstain from responding to a given question to prevent incorrect answers. SEMAN-TICDPP uses determinantal point processes (DPPs) to learn a model over semantic (dis)similarity in model embeddings, which we use in a fully unsupervised manner to identify semantically distinct sets of responses. We additionally present an extension, SEMANTICDPP-C, which yields a soft clustering of the sets of semantically distinct responses. Our extensive empirical investigation examines the behavior of our methods on two frontier open-sourced models of different capacities (that grant access to model internals), Gemma2 9B and Gemma3 27B, on a broad range of widely used QA benchmarks. SEMANTICDPP enables fast uncertainty quantification while it matches or exceeds the selective prediction (hallucination detection) performance of state-of-the-art baselines.

## 1 INTRODUCTION

Large language models generate text by sampling from a distribution over token sequences conditioned on a prompt. This distribution provides valuable information about the model's uncertainty w.r.t. that prompt, but is challenging to use due to its high dimensionality. Exhaustive enumeration of all possible responses is intractable and simple summaries, like individual token probabilities, often capture superficial variations (syntax, synonyms) rather than semantic differences, i.e. differences in meaning. Our goal is to efficiently summarize the posterior distribution over token sequences as a small set of semantically distinct responses, along with corresponding probabilities. Assuming that semantically similar responses lead to the same downstream decisions, we treat them as equivalent.

Recently, Kuhn et al. (2023) and Farquhar et al. (2024) showed that *Semantic Entropy*, the variability in semantic meaning of sampled model responses to a prompt, is a strong indicator of (non)reliability of a generated response. Through querying another language model to find semantically similar responses, they clustered samples according to their meaning and used the normalized cluster counts as probabilities. They then showed that these probabilities were highly indicative of whether a model could correctly answer a question or was guessing or "hallucinating". While effective, this method's practical application is hampered by the prohibitive cost of requiring a second LM to compare pairs of sampled responses. Furthermore, because hard clustering only considers whether responses are equivalent, ignoring the degree of similarity between them, it inherently provides a coarse-grained measure of uncertainty.

Similarly to their approach, we propose a method to summarize the predictive distribution by identifying semantically distinct responses and representing the distribution as a set of clusters, each with an associated probability. However, a key difference is that we do so directly from the internal activations of the model that is generating the responses. To achieve this, we define a probabilistic model over semantically distinct subsets of responses using a *determinantal point process* (DPP). Intuitively, a DPP is a probability measure over sets of objects that favors diverse sets. In our framing, this diversity is given by the volume spanned by vector-valued representations of model responses in semantically-meaningful vector space. We formally introduce the concept of a DPP along with some useful properties in Section 3.2, and provide a complete description of our model and methods in Section 3 and Section 4.

We note that exciting recent work has also improved on the practicality and the granularity of Semantic Entropy. For instance, Kossen et al. (2024) and Chen et al. (2024) circumvent clustering by estimating entropy directly from model activations using linear regression and a kernel method, respectively, leading to more efficient entropy estimation. Furthermore, Nikitin et al. (2024) and Chen et al. (2024) compute a kernel entropy measure over LLM-based semantic scores or embeddings respectively. In this work, we present SEMANTICDPP and SEMANTICDPP-C, which:

1. Summarize a distribution over LLM responses.

2. Efficiently quantify uncertainty—inference time is seconds to fractions of a second.

3. Yield fine-grained uncertainty quantification—which we show performs strongly relative to competitive baselines on hallucination mitigation question answering (QA) tasks.

4. Exhibit promising robustness to task domains—ablations on SEMANTICDPP fit on a given QA task demonstrate nontrivial task generalization behavior on other QA tasks.

Our empirical evaluation consists of several widely used baseline methods and important QA tasks. In alignment with evaluations used in the literature (Farquhar et al., 2024), we frame our experiments as using uncertainty for selective prediction. Specifically, the model abstains from answering questions for which its internal uncertainty is above some threshold. By varying the threshold, we trace out precision-recall and ROC curves and report the area under the ROC curve (AUROC). This setting can be viewed as *hallucination detection* or *hallucination mitigation*, under the notion that a model is "hallucinating" (confabulating, fabricating, etc.) answers when model uncertainty is high. Our two methods, SEMANTICDPP and SEMANTICDPP-C outperform or achieve highly competitive performance with extremely efficient inference times.

## 2 BACKGROUND AND RELATED WORK ON SEMANTIC UNCERTAINTY

Accurately quantifying model uncertainty is an important priority in building machine learning and AI systems that are explainable, transparent, and reliable (Ghahramani, 2015). With the rise of LLMs, it is of particular interest to be able to describe, quantify, measure, and reason over model uncertainty conveyed in natural language (Wang et al., 2025; Rudner & Toner).

Prior work in computational linguistics has used probabilistic models to identify uncertain statements in text as a component of tasks like question-answering and reasoning (Jean et al., 2016). This research has progressed from coarse-level detection to recognizing and categorizing specific subtypes of semantic uncertainty (Szarvas et al., 2012). Furthermore, work by Xiao & Wang (2019) has demonstrated that explicitly modeling uncertainty is crucial not only for quantifying model confidence but also for improving performance on NLP tasks such as sentiment analysis and language modeling.

Recent work on semantic uncertainty within LLMs has primarily focused on question answering (QA) tasks, where it is imperative to know a model's confidence to determine whether its output is reliable, and where the correspondence between confidence and accuracy is measurable. Kadavath et al. (2022) introduced a simple method, "P(True)", to study whether language models can self-evaluate the validity of their own claims, using this to predict the probability "P(True)" that their answers are correct. Kuhn et al. (2023) highlights how "semantic equivalence" (i.e. when two phrases have the same meaning) makes uncertainty estimation challenging, and introduce "semantic entropy" as an unsupervised means of predicting model accuracy, using a separate language model to assess semantic similarity between responses using *bidirectional entailment* (described in Section 5.2). Farquhar et al. (2024) provides an empirical expansion on this work, demonstrating the effectiveness

of semantic entropy for detecting incorrect answers (and abstaining from answering when the risk of confabulating is high) within QA tasks.

Furthermore, Kossen et al. (2024) develops an efficient approximation by using a logistic regression probe to learn a map from model embeddings to uncertainty prediction. Nikitin et al. (2024) addresses the limitation of expressivity by developing a kernel entropy measure, where they use LLMs to score similarity between sequences and then construct a positive definite kernel over sequences. While this approach provides a more fine-grained estimation of uncertainty, it remains computationally expensive requiring $O(n^2)$ LLM queries to compute sequence similarities. Chen et al. (2024) further explores the use of kernel methods for estimating uncertainty, where they fit a kernel over an LLM's embeddings and use the log-determinant as a measure of uncertainty.

Qiu & Miikkulainen (2024) estimates uncertainty by using a kernel density estimate to compute semantic variability between responses. They first use a diverse beam search to induce diversity in responses, then compute pairwise bidirectional entailment using another (possibly cheaper) language model and fit the kernel density estimate to compute probabilities. This approach seems complementary to ours, as the DPP provides a distance function on embeddings that would replace the use of another LLM to compute similarity of responses. The work of Shrivastava et al. (2025) proposes a notion of *relative confidence estimation*, whereby they query to ask "Which question are you more confident in answering correctly?" in relation to a pair of questions, which empirically performs well on a variety of QA tasks.

## 2.1 THE EFFICIENCY OF SEMANTIC CLUSTERING USING LANGUAGE MODELS

Much previous work relies on estimating semantic equivalence with a semantic clustering step. In both Kuhn et al. (2023) and Farquhar et al. (2024), the authors suggest using a comparatively smaller 1.5B-parameter DeBERTa-large model for the bidirectional entailment comparison to make semantic clustering faster and more efficient. However, Farquhar et al. (2024) note that they opt instead to use GPT-3.5, since it performed better. Kuhn et al. (2023) also present a clustering algorithm that is far more efficient in the average case when the number of clusters is generally small. This algorithm, also known as the *Leader algorithm* (Hartigan, 1975), initializes the first response as a cluster and then proceeds by (1) iteratively comparing each remaining response to each cluster, (2) adding that response to a cluster if they are similar and, (3) if no clusters are similar, creating a new cluster from that response. The worst case complexity is $O(m^2)$ if all $m$ responses are dissimilar, but only $2mc$ comparisons are necessary, where $c$ is the number of clusters, which is quite favorable if $c \ll m$. However, whereas a naive $O(n^2)$ approach can be trivially parallelized, the Leader algorithm necessitates sequential computation because the clusters are built up iteratively. Therefore, the Leader algorithm requires minimum $m$ sequential language model generations and worst-case $m^2$, which cannot rely on batching. Given the average case performance improvements over naive clustering, we rely on this algorithm to construct our training data, as described in Section 5.2.

# 3 MODELING SEMANTIC DIVERSITY WITH DPPS

## 3.1 REPRESENTING SEMANTICALLY DISTINCT RESPONSES

We consider two sequences of text to be semantically similar if they convey the same meaning (irrespective of their structure, length, or other syntactic elements), with semantic similarity capturing the degree of resemblance of meaning between the texts. We will assume that semantic meaning is represented within the model embeddings, and we will use this to identify responses from an LLM which are semantically distinct. The challenge is to define a distance metric in embedding space that captures semantic similarity while ignoring *syntactic variability*—irrelevant variations including synonyms, syntactic differences, etc.

To measure semantic similarity, we first define a function $\phi(x) : \mathbb{R}^d \to \mathbb{R}^p$ that maps a $d$-dimensional embedding vector $x$, corresponding to a response from a model, to a $p < d$-dimensional semantically meaningful subspace. Then to compute the similarity between two embedding vectors $x_i$ and $x_j$, we define a kernel or covariance function $k(x_i, x_j)$ that depends on distances in this subspace, i.e.,

$$k(x_i, x_j) = \exp(-\alpha \left\| \phi(x_i) - \phi(x_j) \right\|^2) \tag{1}$$

Intuitively, this provides a mapping to a semantically relevant subspace of the model embeddings and measures distances between embeddings in that space. In this work, we take $\phi(x)$ to be a single layer neural network. In the following we explain how we use a Determinantal Point Process to give a tractable likelihood function that allows us to learn the parameters of this covariance function.

## 3.2 LEARNING WITH DETERMINANTAL POINT PROCESSES

To learn a kernel that captures semantic diversity, we build a model using a *determinantal point process* (DPP). A DPP assigns probabilities to subsets of responses $\mathcal{Y}$ from a ground set $\mathcal{V}$, where more diverse subsets $\mathcal{Y}$ are assigned higher probabilities. We train the DPP to model data collected as follows: for each prompt, we sample a set of responses $\mathcal{V}$, perform semantic clustering on $\mathcal{V}$, then generate observed subsets $\mathcal{Y}$ by selecting a random response from each semantic cluster. We train the model to maximize the likelihood of $\mathcal{Y}$ across the set of prompts.

With a DPP, diversity is represented by a kernel function, and the probability of a set is the determinant of the kernel matrix corresponding to a realized subset $\mathcal{Y}$. The DPP is particularly appealing due to its efficient inference properties, including sampling and computation of marginal probabilities.

DPPs were first developed to describe the physical positions of subatomic particles with repulsive properties, as stochastic point processes (Macchi, 1975). They have since been studied extensively in various subfields and applications of matrix theory (Borodin & Rains, 2005; Borodin & Olshanski, 2000), and developed as models of diversity in machine learning (Kulesza & Taskar, 2012)[1].

**Definition 3.1** (Determinantal point process). Given a ground set of objects or data $\mathcal{V}$, a point process $\mathcal{P}$ on $\mathcal{V}$ (denoted by $\mathcal{V}_\mathcal{P}$) is a probability measure on the power set of $\mathcal{V}$ (i.e. $2^\mathcal{V}$). $\mathcal{P}$ is a determinantal point process if for every subset $\mathcal{S} \subseteq \mathcal{V}$, the probability that $\mathcal{S}$ is contained in a random set $\mathcal{Y}$ is $P(\mathcal{S} \subseteq \mathcal{Y}) = \det(K_\mathcal{S})$, where $K \in \mathbb{R}^{n \times n}$ is a symmetric positive-semidefinite matrix indexed by the elements of $\mathcal{V}$. We denote by $K_\mathcal{S}$ the principal submatrix of $K$ obtained by restricting rows and columns indexed in $\mathcal{S}$.

The assumptions on the matrix $K$ are that its eigenvalues must lie between $[0, 1]$ and the marginal probability of any element $e_i \in \mathcal{V}$ is given by $K_{ii}$ ($K$ is often referred to as the *marginal kernel* because it encodes all the information needed to compute the probability of any subset $\mathcal{S}$ being included in $\mathcal{Y}$, i.e. $P(\mathcal{S} \subseteq \mathcal{Y})$). The marginal probability of including any two elements, e.g. $e_i$ and $e_j$, is given by

$$P(\{e_i, e_j\} \subseteq \mathcal{Y}) = K_{ii}K_{jj} - K_{ij}^2 = P(e_i \in \mathcal{Y})P(e_j \in \mathcal{Y}) - K_{ij}^2.$$

Intuitively, the larger the value of $K_{ij}$, the more *unlikely* $e_i$ and $e_j$ are to appear together, which provides a simple model of repulsion between two elements. As such, DPPs model *negative correlations* between points, a useful feature that we leverage in our method.

We employ a particularly useful class of DPPs known as *L-ensembles* (Borodin & Rains, 2005). An L-ensemble is defined by a real, symmetric positive semi-definite matrix $L$ indexed by the elements of $\mathcal{V}$. Within an L-ensemble, the probability of a subset $\mathcal{Y} \subseteq \mathcal{V}$ is proportional to the $\det(L_\mathcal{Y})$, given by

$$P_L(\mathcal{Y}) = \frac{\det(L_\mathcal{Y})}{\det(L + I)}$$

where $L$ is the kernel matrix over the full set $\mathcal{V}$, constructed using our kernel $k(x_i, x_j)$, $L_\mathcal{Y}$ is the submatrix corresponding to the subset $\mathcal{Y}$, and $I$ is the $n \times n$ identity matrix.

While both general DPPs and L-ensembles allow us to compute marginal probabilities, L-ensembles allow us to directly calculate the probability of any selecting any subset $P(\mathcal{S} = \mathcal{Y})$, including the empty set. We capitalize on this loss function and use it as a target within our optimization problem, giving rise to the following loss function:

$$\log P_L(\mathcal{Y}) = \log \det(L_\mathcal{Y}) - \log \det(L + I) \tag{2}$$

Optimizing this loss seeks to find a kernel $L$ that maximizes the determinant (diversity) of subset $\mathcal{Y}$ via the first term, while minimizing the determinant of all possible subsets via the second term,

---

[1]We provide a high-level overview of the most relevant concepts of DPPs in this work to provide the reader with necessary background and intuition to motivate and understand our methodology, and refer the reader to the prior work (Kulesza & Taskar, 2012) for further in-depth exploration of this elegant probabilistic model.

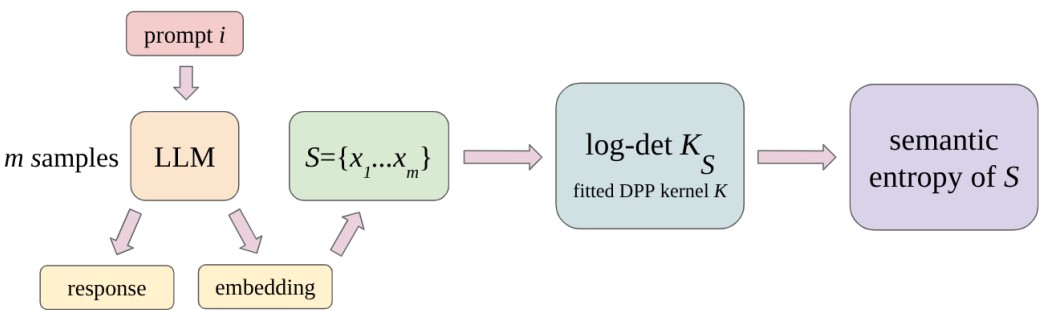

Figure 1: A diagram of SEMANTICDPP. For a given prompt $i$, we sample $m$ responses from an LLM and extract the model embeddings corresponding to each response. We represent these $m$ responses as a matrix, $\mathcal{S}$, with dimension $m \times$ embedding_dim. Then, using the fitted DPP's marginal kernel $K$, we obtain the entropy of $\mathcal{S}$ by computing the log-determinant of $K_{\mathcal{S}}$ (the submatrix $K_{\mathcal{S}}$ w.r.t. $\mathcal{S}$), given by Equation (4).

thus making $\mathcal{Y}$ more likely relative to all other options. Intuitively, if one uses a stationary kernel with Euclidean distances on some latent space, as we do, then optimizing Equation 2 resembles contrastive learning. The first term $\log \det(L_{\mathcal{Y}})$ pushes objects in $L_{\mathcal{Y}}$ apart in latent space, while the second term $\log \det(L + I)$ pulls everything closer together. We find that the second term provides effective regularization, preventing the need for any additional regularization terms such as priors on the parameters of the kernel.

## 4 METHODS

In this section, we present two new algorithms, SEMANTICDPP, and a clustering-based extension, SEMANTICDPP-C. Both methods use the following core model: for given dataset of $n$ prompts and an LLM, we first fit a DPP under the assumption that we have a collection of semantically distinct subsets $\mathcal{Y} = \{\mathcal{Y}_i\}$ responses, where $\mathcal{Y}_i$ denotes the subsets corresponding to each prompt $i$. Details of how we obtain this in our empirical evaluation is described in Section 5.2. Note that the DPP model is fit *only once*, i.e. inference is amortized on a training set.

**Response embedding** Our algorithm requires a single vectorized or embedding representation for each response. There are multiple strategies one could use in representing the generating LLM's internal state for a given response, e.g. extract the embedding vector from the last layer of the LLM, extract the embedding for each generated token and combining them in some manner, etc. We discuss the choice we make in our empirical evaluation in Appendix C.

**Fitting the DPP** To fit the DPP, we use the semantically distinct subsets $\mathcal{Y}_i$ of data. For each given prompt $i$, we sample a single response from each of the semantically distinct subsets $\mathcal{Y}_i$, giving rise to $n$ pseudolabeled subsets of responses. We use the loss function we derive in Equation (2) and minimize the average negative marginal log likelihood ($\mathcal{NLL}$) under the responses, given by

$$\mathcal{NLL} = -\frac{1}{n} \sum_{i=1}^{n} \log \det(L_{\mathcal{Y}_i}) - \log \det(L_i + I) \tag{3}$$

where $L_{\mathcal{Y}_i}$ is the submatrix indexed by the subset of response indices $\mathcal{Y}_i$ and $L_i$ is the kernel over all responses for prompt $i$.

### 4.1 SEMANTICDPP

SEMANTICDPP estimates the uncertainty corresponding to any set $S$ of responses to a prompt $i$ as the log probability of the set under our learned DPP with marginal kernel $K$. Recall from Definition 3.1 that we can use $K$ to compute the marginal probability of any subset. We first represent the collection of $m$ output sequences as a set of embeddings $\mathcal{S}$ (with dimension $m \times$ embedding_dim) and then compute our $m \times m$-dimensional kernel $K_{\mathcal{S}}$ by applying Equation (1) pairwise. We then compute

the log-determinant of the kernel sub-matrix corresponding to $\mathcal{S}$, given by

$$\log P(\mathcal{S}) = \log \det K_{\mathcal{S}} = \log \det L_{\mathcal{S}}(L_{\mathcal{S}} + I)^{-1}, \tag{4}$$

where $L_{\mathcal{S}}$ is the principal submatrix of $L$ w.r.t. $\mathcal{S}$. To help build intuition for the method and illustrate on how SEMANTICDPP compares with Semantic Entropy, Figure 1 provides a workflow of the process of producing the semantic entropy of a prompt $i$ for a given LLM.

### 4.2 SEMANTICDPP-C

In SEMANTICDPP-C, we use the DPP machinery to cluster a set of sampled responses $\mathcal{Y}$ into semantically distinct clusters, and assign each a probability. To achieve this, we need to find the subset of responses $S' \subseteq \mathcal{Y}$ that maximizes the probability under the DPP, proportional to $det(L'_S)$, as these constitute the distinct clusters. The remaining responses, $\mathcal{Y} \setminus S'$, can then easily be assigned to most similar cluster using the pairwise similarity given by $K$.

Finding the most likely subset (the mode) of a DPP is NP-hard. However, two properties of DPPs enable an efficient greedy approximation: (1) the log-determinant is a submodular function, making it suitable for greedy approximation algorithms and (2) the expected cardinality of subsets sampled under the DPP is analytically tractable.

---

**Algorithm 1** SEMANTICDPP-C

1: $C = \{x_0\}$ {subset of cluster "centers"}
2: $S = \{x_1...x_m\}$ {remaining samples (responses)}
3: $K = L(L+I)^{-1}$ {compute the DPP marginal kernel}
4: $c = \max(1, \mathrm{round}(\mathrm{tr}(K)))$ {estimate the expected cardinality}
5: **for** $i = 1$ to $c$ **do**
6: $\quad C \leftarrow C \cup \arg\max_j \log \det(L_{C \cup S_j})$
7: $\quad S \leftarrow S \setminus I$ {remaining indices}
8: **end for**
9: **for** $i \leftarrow 1$ to $|C|$ **do**
10: $\quad \mathrm{clusters}_i = \{C_i\}$
11: **end for**
12: **for** $i$ in $|S|$ **do**
13: $\quad$ {assign to most similar cluster center.}
14: $\quad j = \arg\max L_{i,C}$
15: $\quad \mathrm{clusters}_j = \mathrm{clusters}_j \cup S_i$
16: **end for**

---

As a first step in our approximation, we estimate the target number of clusters, $k$. Given a subset $\mathcal{S} \subseteq \mathcal{Y}$, if $\mathcal{S}$ is sampled according to the DPP's marginal kernel matrix $K$, then the expected cardinality of $\mathcal{S}$ is given by trace($K$), i.e. $\mathbb{E}|S| = tr(K)$. We use this to set the target size of our cluster set.

Given the cardinality, $k$, there are several approaches to approximately find the maximum *a posteriori* (MAP) subset of size $k$. We use the method *Greedy MAP approximation* from Kulesza & Taskar (2012) which builds upon the approach of Çivril & Magdon-Ismail (2009) used to find maximum-volume submatrices (another way of viewing this problem).

We use the expected cardinality of $S'$ to greedily add responses to construct the set $S'$ approximating

$$S'_{MAP} = \underset{S' \subseteq \mathcal{Y}, |S'| = k}{\arg\max} \det(L_{S'}).$$

In practice, we set $k = \max(1, \mathrm{round}(\mathrm{trace}(K)))$ to ensure the target cardinality is a positive integer. Our algorithm is detailed in Algorithm 1, where responses representing clusters are denoted by $C$.

## 5 EXPERIMENTAL EVALUATION

To assess the efficacy of our approach to measure uncertainty and apply this to important downstream tasks, we frame our empirical evaluation as "selective prediction" for confabulation ("hallucination", fabrication, etc.) mitigation within question-answering tasks, by allowing the model to abstain from answering a question if the estimated uncertainty exceeds a certain threshold. By varying this threshold we can trace out precision-recall curves, receiver-operator characteristic (ROC) curves and compute the Area Under the ROC (AUROC) metric to quantitatively compare methods. While our approach is language agnostic, we focus on the following datasets as empirical evidence, acknowledging that they are all in English (refer to Section 6 for more detail).

**Datasets** We conduct experiments on six widely used question answering (QA) benchmark datasets, selected to have overlap with existing literature and cover different challenges within QA tasks: TruthfulQA (Lin et al., 2022), GSM8k (Cobbe et al., 2021), TriviaQA (Joshi et al., 2017), BioASQ (Tsatsaronis et al., 2015), NQ Open (Lee et al., 2019), and Squad (Rajpurkar et al., 2016). We provide an overview, including version details, for reproducibility in Appendix C.1.

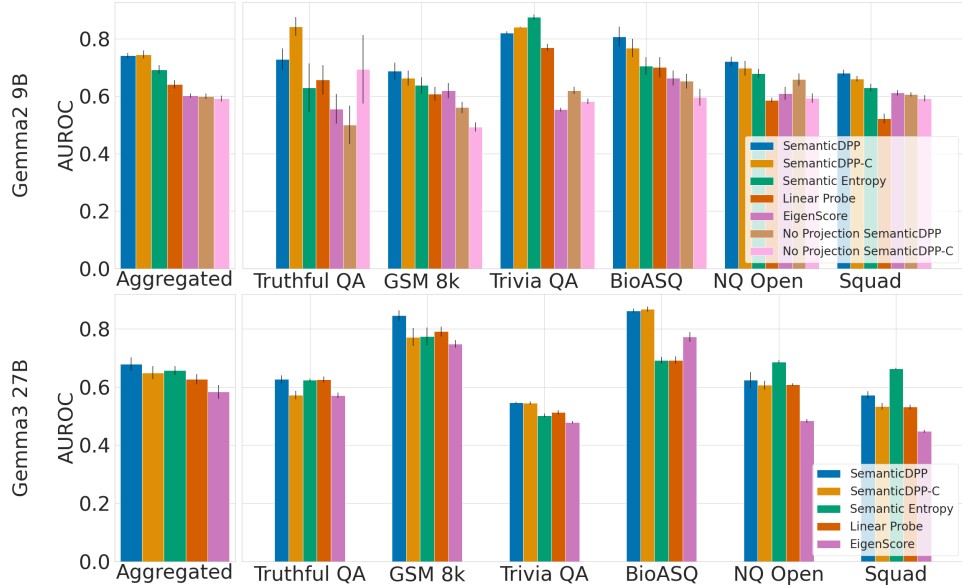

Figure 2: Comparative AUROC performance of SEMANTICDPP and SEMANTICDPP-C against several baselines—Semantic Entropy, Linear Probe, EigenScore, and our two no-projection baselines. Depicted is the mean AUROC across 5 random train/test folds and the error bars indicate the standard deviation using Gemma2 9B and Gemma3 27B to generate responses.

## 5.1 BASELINES

We compare SEMANTICDPP and SEMANTICDPP-C with several baselines of various complexities to evaluate their performance across the aforementioned datasets. We also evaluate on two simplified ablations of SEMANTICDPP and SEMANTICDPP-C.

**Semantic Entropy** (Farquhar et al., 2024), relies on a computationally intensive, bidirectional entailment-based clustering approach (Section 5.2). Clusters are assigned probabilities by counting the number of responses in each cluster and normalizing. Standard Shannon-entropy is computed from these probabilities.

**Linear Probe** (Kossen et al., 2024) involves fitting a linear regression from the averaged and whitened embedding representation of a single response to the entropy as computed in Semantic Entropy.

**EigenScore** (Chen et al., 2024, Eq. 5) computes the log-determinant of a centered linear kernel on the same whitened embeddings as SEMANTICDPP.

**No Projection SEMANTICDPP/SEMANTICDPP-C** each use the corresponding base algorithm, but fits the DPP model directly on response embeddings, taking $\phi$ to be the identity rather than a single layer NN with learned parameters, and using kernel function (Equation (1)) to optimize the hyperparameters $\alpha$ and the scaling parameters for each dimension.

**P(True)** (Kadavath et al., 2022) is a single-response method that predicts "P(True)" that the model's response is correct, using a logistic regression head.

**Token Likelihoods** is a single-response method that uses the average (length-normalized sum) of the individual output token log-likelihoods w.r.t. a single response (the log perplexity of the response).

## 5.2 EXPERIMENTAL SETUP

Here we describe how we produce training data for fitting the DPP used in SEMAN-TICDPP and SEMANTICDPP-C. A detailed overview for reproducibility can be found in Appendix C.

**Obtaining semantically distinct subsets to fit the DPP** As a proxy for "ground truth" semantically distinct subsets of responses with which to fit our DPP model, we use the approach from Farquhar et al. (2024) to cluster the data and use each response's cluster assignment to pseudolabel this data. We optimize the marginal likelihood of the DPP conditioned on these labeled subsets. Note that other approaches may be used to obtain semantically distinct responses, which would make for an interesting ablation in follow-up work.

Table 1: AUROC Scores of Different Methods with Gemma2 9B

| Method | Aggregated | Truthful QA | GSM 8k | Trivia QA | BioASQ | NQ Open | Squad |
|---|---|---|---|---|---|---|---|
| SEMANTICDPP | $\mathbf{0.74 \pm 0.01}$ | $0.73 \pm 0.04$ | $\mathbf{0.69 \pm 0.03}$ | $0.82 \pm 0.00$ | $\mathbf{0.81 \pm 0.03}$ | $\mathbf{0.72 \pm 0.02}$ | $\mathbf{0.68 \pm 0.01}$ |
| SEMANTICDPP-C | $\mathbf{0.74 \pm 0.01}$ | $\mathbf{0.84 \pm 0.03}$ | $0.66 \pm 0.03$ | $0.84 \pm 0.00$ | $0.77 \pm 0.03$ | $0.70 \pm 0.03$ | $0.66 \pm 0.01$ |
| EigenScore | $0.60 \pm 0.01$ | $0.56 \pm 0.05$ | $0.62 \pm 0.03$ | $0.55 \pm 0.01$ | $0.66 \pm 0.03$ | $0.61 \pm 0.02$ | $0.61 \pm 0.01$ |
| Semantic Entropy | $0.69 \pm 0.02$ | $0.63 \pm 0.08$ | $0.64 \pm 0.03$ | $\mathbf{0.87 \pm 0.01}$ | $0.70 \pm 0.03$ | $0.68 \pm 0.02$ | $0.63 \pm 0.01$ |
| Linear Probe | $0.64 \pm 0.02$ | $0.66 \pm 0.05$ | $0.61 \pm 0.02$ | $0.77 \pm 0.01$ | $0.70 \pm 0.03$ | $0.59 \pm 0.01$ | $0.52 \pm 0.01$ |
| No Proj SEMANTICDPP | $0.60 \pm 0.01$ | $0.50 \pm 0.07$ | $0.56 \pm 0.02$ | $0.62 \pm 0.01$ | $0.65 \pm 0.03$ | $0.66 \pm 0.02$ | $0.61 \pm 0.01$ |
| No Proj SEMANTICDPP-C | $0.59 \pm 0.01$ | $0.69 \pm 0.12$ | $0.49 \pm 0.02$ | $0.58 \pm 0.01$ | $0.60 \pm 0.03$ | $0.59 \pm 0.02$ | $0.59 \pm 0.01$ |
| P(True) | $0.60 \pm 0.02$ | $0.50 \pm 0.02$ | $0.62 \pm 0.04$ | $0.53 \pm 0.01$ | $0.80 \pm 0.06$ | $0.66 \pm 0.02$ | $0.50 \pm 0.01$ |
| Token Average Likelihoods | $0.56 \pm 0.02$ | $0.29 \pm 0.03$ | $0.64 \pm 0.01$ | $0.52 \pm 0.02$ | $0.69 \pm 0.07$ | $0.63 \pm 0.01$ | $0.59 \pm 0.01$ |

Table 2: AUROC Scores of Different Methods with Gemma3 27B

| Method | Aggregated | Truthful QA | GSM 8k | Trivia QA | BioASQ | NQ Open | Squad |
|---|---|---|---|---|---|---|---|
| SEMANTICDPP | $\mathbf{0.68 \pm 0.02}$ | $\mathbf{0.63 \pm 0.01}$ | $\mathbf{0.85 \pm 0.02}$ | $0.55 \pm 0.00$ | $0.86 \pm 0.01$ | $0.62 \pm 0.03$ | $0.57 \pm 0.01$ |
| SEMANTICDPP-C | $0.65 \pm 0.02$ | $0.57 \pm 0.01$ | $0.77 \pm 0.03$ | $\mathbf{0.55 \pm 0.01}$ | $\mathbf{0.87 \pm 0.01}$ | $0.61 \pm 0.02$ | $0.53 \pm 0.01$ |
| EigenScore | $0.58 \pm 0.02$ | $0.57 \pm 0.01$ | $0.75 \pm 0.01$ | $0.48 \pm 0.00$ | $0.77 \pm 0.02$ | $0.48 \pm 0.01$ | $0.45 \pm 0.01$ |
| Semantic Entropy | $0.66 \pm 0.02$ | $0.62 \pm 0.00$ | $0.77 \pm 0.03$ | $0.50 \pm 0.01$ | $0.69 \pm 0.01$ | $\mathbf{0.69 \pm 0.01}$ | $\mathbf{0.66 \pm 0.00}$ |
| Linear Probe | $0.63 \pm 0.02$ | $0.63 \pm 0.01$ | $0.79 \pm 0.02$ | $0.51 \pm 0.01$ | $0.69 \pm 0.01$ | $0.61 \pm 0.00$ | $0.53 \pm 0.01$ |

For a given data set of $n$ prompts, we sample $m$ responses for each from an LLM. We then cluster these responses using *pairwise bidirectional entailment*–for two statements $s_a, s_b$, $s_a$ and $s_b$ entail one another if and only if they both imply each other. Using this as our similarity and cluster membership condition, we add elements to the same cluster when they entail each other via the more computationally efficient Leader algorithm (as detailed in Section 2.1). This step yields a collection of clusters assignments for each prompt, $C := C_0, ..., C_n$, which we use to psuedolabel our responses to serve as a proxy for ground truth semantic similarity. In our experiments, we use the publicly available Gemini 1.5 Flash model to produce this pseudolabeling (Google Gemini Team et al., 2024).

**Model choices** Our empirical evaluation focuses on evaluating our methods and baselines on two powerful, instruction-tuned, open-source LLMs with different model capacities that allow users to access model internals: Gemma2 9B (Google Gemma Team et al., 2024) and Gemma3 27B[2]. For each benchmark dataset, we create 75%-25% train/test splits and select the model hyperparameters with the best marginal likelihood on the training set (refer to Table 5 and Table 6 for a full breakdown).

### 5.3 EMPIRICAL RESULTS

In Figure 2, we present AUROC scores with error bars for the most competitive methods on each benchmark and generating LLM, along with the average (aggregated) AUROC across datasets. In Appendix A, we present Precision-Recall (PR) and ROC curves ( Figure 3) to highlight the top uncertainty method's performance as a function of the cutoff threshold at which to abstain from responding. As we remark in Appendix C.2, it's worth noting that at least two of these widely employed benchmark datasets (TriviaQA and NQ Open) include questions which rely on an assumed temporal context, and exhibit temporal shift—as in, the labeled answer was true when the benchmark was created, but is has potentially become untrue in the time that's passed. We wish to bring this to the attention of benchmark curators to drive intentional benchmark design considerations.

|  | Fit on | | |
|---|---|---|---|
| Transfer to | BioASQ | GSM8k | TruthfulQA |
| GSM8k | 0.82 | 0.85 | 0.82 |
| NQ-Open | 0.56 | 0.55 | 0.58 |
| Squad | 0.60 | 0.58 | 0.60 |
| TruthfulQA | 0.63 | 0.58 | 0.63 |
| TriviaQA | 0.53 | 0.52 | 0.55 |
| BioASQ | 0.86 | 0.60 | 0.65 |
| Avg. score | 0.658 | 0.613 | 0.655 |

Table 3: Task transfer performance (AUROC) of SEMANTICDPP using Gemma3 27B.

Across benchmark tasks, SEMANTICDPPand SEMANTICDPP-C are top scoring methods in terms of AUROC using both Gemma2 9B and the larger capacity model, Gemma3 27B, as the generating LLM. In aggregate, SEMANTICDPP is the strongest performing model across tasks for each model.

On NQ-Open and Squad with Gemma3 27B: upon manual inspection of many high-confidence but incorrect answers from both SEMANTICDPP and  SEMANTICDPP-C, we observe that many

---

[2]Gemma3 27B "delivers state-of-the-art performance for its size, outperforming Llama3-405B, DeepSeek-V3 and o3-mini in preliminary human preference evaluations on LMArena's leaderboard" (The Keyword, Mar 2025)

examples had incorrect ground-truth answers. As with the prior note about temporal distribution shift, we imagine that errors in the benchmark itself—either due to answers or facts changing with time, or QA answers simply being incorrect—that this has a more pronounced impact when using a generating model that either (1) was trained more recently, or (2) has a larger, richer, and more factual internal knowledge/representation.

To test how well our model generalizes across tasks, we conducted a transfer experiment, training the DPP on one dataset and evaluating the AUROC of SEMANTICDPP on all others (see Table 3). A key concern was that the learned kernel might be too task-specific; for example, a model trained on GSM8k might only learn to represent similarity for mathematical answers. Indeed, we found that training on GSM8k led to a significant drop in performance on other tasks, which is unsurprising given its highly structured answers. However, models trained on broader datasets like BioASQ and TruthfulQA generalized well, with only a slight decrease in the aggregated AUROC score. Interestingly, the model also generalized well to GSM8k when trained on trivia questions, suggesting it learns a broadly applicable measure of semantic distance.

## 5.4 ON COMPUTATIONAL OVERHEAD

In Table 4, we illustrate the practical computational cost comparison of each method on TriviaQA to emphasize the computational efficiency of both SEMANTICDPPand SEMANTICDPP-C. To accommodate Semantic Entropy's compute overhead, these runtimes reflect an implementation efficiency we add by parallelizing the bidirectional entailment computation over 16 replicated Gemini 1.5 Flash model instances (1,433,553 total comparisons). We additionally provide the approximate computational complexity of our methods along with the baselines we compare against in Table 7 in Appendix E.

| Method | Runtime |
|---|---|
| SEMANTICDPP | 0.00186s |
| SEMANTICDPP-C | 29.6s |
| Semantic Entropy | 12914s x 16 ≈ 57.4hrs |
| Linear Probe | 4.8914s |
| EigenScore | 0.00148s |

Table 4: Wall clock comparison of methods on TriviaQA, using one fold of evaluation on Gemma2 9B. Runtime is the actual **cumulative** compute time over the entire test split of 9,961 examples.

## 6 LIMITATIONS

While our approach is language agnostic, we focus on a collection of widely employed benchmark datasets to validate our method, which are all in English: a future direction is to explore this technique in the multilingual setting. Our method also still requires sampling multiple responses from an LLM in order to quantify semantic variability. Thus we incur a cost relative to the number of samples in order to quantify uncertainty. Moreover, we require access to model embeddings. While using an LLM to do semantic clustering is significantly more expensive, it is more flexible and transferable.

We assume that the distribution over responses is quite peaked around a small number of clusters. However, often one is interested in *tail probabilities* or the probability of unlikely but high risk/reward responses such that the expected value of a downstream decision could be significant. Currently such responses are unlikely to be captured by our method, as sampling them is quite unlikely. An interesting future direction would be to estimate the likelihood of such sequences directly based on distances in the embeddings.

## 7 CONCLUSION

We present SEMANTICDPP, a novel method for efficiently estimating model uncertainty using model embeddings corresponding to LLM responses for any prompt, and its extension, SEMANTICDPP-C. These methods leverage determinantal point processes (DPPs), which allow us to compute the probability of the set of responses to a given prompt under a DPP, yielding a measure of the diversity of the set. Within SEMANTICDPP-C, we use the same DPP base model to identify semantically distinct sets of responses and assign each a probability. While we center our experimental validation on the important downstream task of hallucination mitigation, the accurate quantification of uncertainty is important for tasks such as decision making, out-of-distribution generalization, and event forecasting. We show that both SEMANTICDPP and SEMANTICDPP-C achieve high performance on various important benchmarks compared against several popular, widely used baselines—and does so with relatively minimal additional overhead.

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

## A  ADDITIONAL FIGURES: ROC AND PR CURVES

In Figure 3, we present Precision-Recall and ROC curves, which highlight the top uncertainty methods' performance as a function of the cutoff threshold at which to abstain from responding.

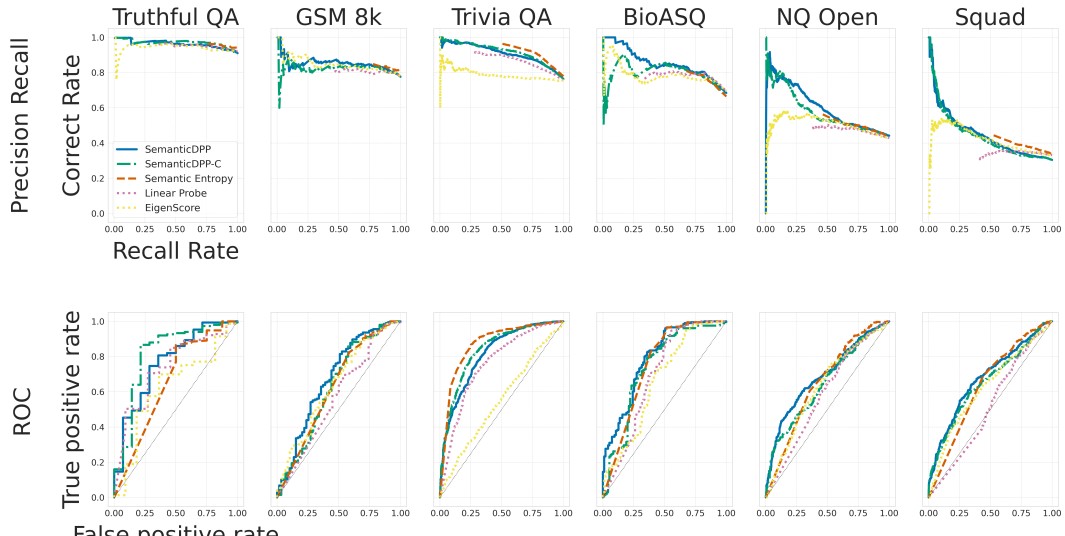

Figure 3: Precision-recall (top) and ROC (bottom) curves for SEMANTICDPP, SEMANTICDPP-C and baselines on the six benchmarks with Gemma2 9B for one random fold (i.e. seed 0). Note that, for the PR curves, some methods start with a recall-rate well above 0. By using distances in the embedding space, our methods give more fine-grained uncertainty at high precision than Semantic Entropy's clustering. Similarly reflected in the ROC curves, SEMANTICDPP and SEMANTICDPP-C outperform Semantic Entropy at low-false positive rates. Interestingly, embedding-based methods can be wrong with very high confidence (i.e. far left of the PR curve). We found that with e.g. NQ Open this is due to many examples having incorrect answers, because they are time-based and the actual answer has since changed.

## B  MORE DETAILS ON FITTING A DPP

In learning (fitting) the parameters of a DPP in our experiments, we use a Gaussian prior on the parameters of the low-rank projection in the covariance matrix. We maximize the marginal likelihood of the DPP loss function (Equation (2)) using a batched gradient descent method (Adam), sampling random cluster centers from each cluster and taking a gradient step to maximize the marginal likelihood of that set.

## C  EXPERIMENTAL DETAILS FOR REPRODUCIBILITY

**Hyperparameters** We performed a hyperparameter search to select hyperparameters for each model on each dataset. This was done by selecting the hyperparameters that gave the highest DPP marginal log-likelihood on a separate held out validation set. In Table 5 and Table 6 we show the hyperparameters that were selected for Gemini 2 9B and Gemini 3 27B respectively. The search spaces are provided in the second row. For Gemma 2 27B we performed mini-batch gradient updates due to the higher memory requirements of the larger model embeddings and searched over the batch size.

**Kernel function** Our kernel function is inspired by the Mahalanobis distance function, $d(x_i, x_j) = \sqrt{(x_i - x_j)^T M^{-1}(x_i - x_j)}$, where $M$ is a positive semi-definite matrix providing a linear projection onto a lower dimensional subspace. However, for the tasks we explore in our experimental evaluation, a linear projection $M$ appears insufficiently expressive to capture the complexity of relationships between datapoints when used in our kernel $k$ (Equation (1)). In light of this, we extend this to

Table 5: Hyperparameter settings chosen according to highest marginal log-likelihood on the validation set for Gemma 2 9B.

| Dataset | Learning rate (0.0001–0.02) | Hidden units [25, 50, 150, 1000] | Low rank dim [2, 5, 10, 30] | Batch Size - |
|---------|-----------------------------|----------------------------------|-----------------------------|--------------|
| Truthful QA | 0.003987 | 150 | 30 | Full Dataset |
| GSM 8k | 0.008881 | 150 | 100 | Full Dataset |
| Trivia QA | 0.004903 | 1000 | 100 | Full Dataset |
| BioASQ | 0.008881 | 150 | 100 | Full Dataset |
| NQ Open | 0.005958 | 50 | 10 | Full Dataset |
| Squad | 0.046871 | 150 | 2 | Full Dataset |

Table 6: Hyperparameter settings chosen according to highest marginal log-likelihood on the validation set for Gemma 3 27B.

| Dataset | Learning rate (0.0001–0.02) | Hidden units [25, 50, 150, 1000] | Low rank dim [2, 5, 10, 30] | Batch size [24,64,128,256,1024] |
|---------|-----------------------------|----------------------------------|-----------------------------|---------------------------------|
| Truthful QA | 0.006607 | 25 | 2 | 128 |
| GSM 8k | 0.014223 | 1000 | 5 | 24 |
| Trivia QA | 0.000167 | 25 | 2 | 128 |
| BioASQ | 0.000385 | 1000 | 30 | 128 |
| NQ Open | 0.000599 | 1000 | 5 | 64 |
| Squad | 0.013053 | 1000 | 5 | 128 |

a nonlinear mapping (denoted by $\phi$), and find that the non-linearity offered by a simple one-layer network works well in practice. Thus, We use a single hidden layer neural network as $\phi$, the nonlinear mapping in our kernel function (Equation 1). We used a Gaussian Error Linear Unit (GELU) activation function. We learned the parameters of this projection jointly with the kernel hyperparameters to minimize the negative Marginal Log-Likelihood of the DPP, using the Adam optimizer.

**Embedding representation** For each sampled response, we also extract the embeddings of the last layer of the model, and average these over the generated token sequence. To facilitate the use of the last-layer embeddings as a semantic representation of sequences, we first perform a whitening transformation that is equivalent to the causal inner product of Park et al. (2024), eq 3.3. We found this transform to be universally beneficial, but we expect it to be especially useful to methods that rely on linear transformations of the embeddings (i.e. EigenScore, Linear Probe).

Note that in our experiments using SEMANTICDPP-C, we use the largest cluster as the proposed answer and compute entropy similarly to the Semantic Entropy, i.e. as standard Shannon entropy.

## C.1 BENCHMARK DATASETS

**TruthfulQA** is a factual QA dataset that is carefully crafted to contain questions that reflect common misconceptions and falsehoods. Following standard practice, we use a fine-tuned GPT-3.5 Turbo model, "GPTJudge" Lin et al. (2022), to score each sampled answer from the model.

**GSM8k** is a high-quality grade school math reasoning dataset that is designed to require multi-step reasoning to arrive at a correct numerical answer. To score responses, we perform a regular expression search for the last number in the response and compare to the ground-truth answer.

**TriviaQA** is a dataset of trivia QA pairs authored by trivia enthusiasts. We depart from the standard framing as reading comprehension and instead follow the setup of Farquhar et al. (2024) and treat this as a pure QA dataset, in that we pass the trivia questions as a prompt to the model without any additional context, then sample $m$ responses and score each response. To score responses, we use another LLM, Gemini 1.5 Flash (Google Gemini Team et al., 2024), to compare the response to the provided ground truth answer.

**BioASQ** is a large-scale, multi-year biomedical semantic indexing and question answering challenge. We use the version distributed by TensorFlow Datasets, which is the test set from the MRQA 2019 Shared Task benchmark (Fisch et al., 2019).

**NQ Open 2.0** is a challenging, open-domain question answering task derived from Natural Questions (Kwiatkowski et al., 2019), which contains real user questions issued to Google search and annotated answers found from Wikipedia.

**SQuAD 2.0** (Stanford Question Answering Dataset) is a reading comprehension dataset where questions were produced by crowdworkers regarding Wikipedia articles; answers are either drawn directly from the article or are labeled as unanswerable. We use version 2.0 from TensorFlow Datasets—because the unanswerable questions depend directly on the context provided with the question, we filter the 'impossible' questions from the dataset in our experiments.

## C.2 ERRORS IN DATASETS

Two of our benchmark datasets (Trivia-QA and NQ-Open) include questions which depend on a temporal component. This can lead to **temporal shift**: the answer which was true when the benchmark was created has become stale and is no longer true. In this case, even if a model responds with what is currently the correct answer, it is penalized for not aligning with the answer which was true at the time. This is particularly true when a question is phrased relative to "now" or "the last time" without further specification of time. Additionally, questions may be ambiguously stated with respect to time; for instance, they reference a given month without a year specified.

To provide an illustrating example, consider the following from NQ-Open:

> **question:** "When was the last time the Carolina Hurricanes made the playoffs?"
> **answer:** "2008-2009"
> **correct answer in 2025:** The Hurricanes have been in the playoffs every year since 2019 and are currently in the playoffs.

Going forward, we encourage the temporal aspect to be considered very intentionally in the creation of benchmarks, ensuring that intended answers are robust to distribution shift or are scoped to measure model behavior under temporal ambiguity.

# D    LLM PROMPTS

In alignment with Farquhar et al. (2024), we use the following prompt to obtain a proxy of ground truth semantic cluster labels, used to fit the DPP (Section 4. This LLM-based sequence comparison uses *semantic entailment* as its measure of similarity.

```
"""
We are evaluating answers to the question {question}

Here are two possible answers:

Possible Answer 1: {answer_1}

Possible Answer 2: {answer_2}

Does Possible Answer 1 semantically entail Possible Answer 2? Respond with
A) entailment
B) contradiction
C) neutral

== Letter:
"""
```

| Algorithm | Training (LLM calls) | Prediction (LLM Calls) |
|---|---|---|
| DPP | negligible | $m$ |
| SEMANTICDPP | $n * m^2$ | $m$ |
| SEMANTICDPP-C | $n * m^2$ | $m$ |
| Semantic Entropy | negligible | $m^2$ |
| Linear Probe | $n * m^2$ | 1 |
| EigenScore | negligible | $m$ |

Table 7: Computational complexity estimates of our algorithms and baselines we compare against. Recall that $m$ is the number of sampled LLM responses required per prompt and $n$ is the size of the training dataset. For the methods that require training, the dominant cost is producing training data. Many of the methods choose to use Semantic Entropy to produce training data in their experiments and thus require $m^2$ LLM generations for each of the $n$ examples in the dataset. However, once the methods are trained, the dominant cost is in producing sampled LLM responses to a prompt.

| Dataset | # Examples | SemanticDPP | SemanticDPP-C | EigenScore | Linear Probe | Semantic Entropy (# comparisons) |
|---|---|---|---|---|---|---|
| Truthful QA | 817 | $1.26 \pm 1.03$ | $8.64 \pm 4.95$ | $2.20 \pm 1.96$ | $0.30 \pm 0.09$ | 696.75 (32273) |
| GSM 8k | 1319 | $0.83 \pm 0.74$ | $6.15 \pm 2.03$ | $2.09 \pm 1.87$ | $0.41 \pm 0.13$ | 481.61 (48494) |
| Trivia QA | 9961 | $2.25 \pm 2.01$ | $32.79 \pm 3.79$ | $2.36 \pm 2.11$ | $14.88 \pm 0.13$ | 12638.07 (1430462) |
| BioASQ | 817 | $0.80 \pm 0.71$ | $8.39 \pm 1.98$ | $2.10 \pm 1.88$ | $0.34 \pm 0.10$ | 696.75 (32273) |
| NQ Open | 3610 | $3.30 \pm 2.95$ | $15.65 \pm 6.54$ | $2.16 \pm 1.93$ | $1.30 \pm 0.07$ | 2909.62 (189641) |
| Squad | 5924 | $3.36 \pm 3.01$ | $21.35 \pm 4.39$ | $2.28 \pm 2.04$ | $6.41 \pm 0.18$ | 2428.01 (234698) |

Table 8: Comparison of Method Runtimes (in seconds) Across Datasets. This table shows wall-clock times for uncertainty prediction for each method on each dataset. We summed the wall-time over the whole test-set and report the mean and standard error. For Semantic Entropy we also report the total number of LLM-based bidirectional entailment comparisons were performed, as the timing depends on the LLM being used.

# E  COMPUTATIONAL OVERHEAD

Here we provide theoretical and empirical estimates on the computational complexity of the various methods. In Table 7, to provide greater clarity on the computational requirements for our proposed methods and baselines, we provide a breakdown of complexity in terms of calls to an LLM—the most costly computational component across the methods. Note that in terms of fitting a DPP, the model requires labeled pairs of data, for which we choose to use the Semantic Entropy method to produce these. We remind the reader that this is not the only way to obtain labeled pairs, nor the only choice for producing such data. As such, this data preprocessing cost is not inherent to our approach.

In general, the computational complexity of a DPP is dominated by taking the determinant of the kernel, which is cubic in the size of the set. However, in this application $m^3$ over a real-valued kernel is negligible compared to sampling from an LLM $m^2$ times. Thus in Table 7, we focus on LLM generations (calls), noting that this is a coarse measure.

We also note that the Semantic Entropy and EigenScore approaches rely on pre-trained LLMs, which we do not count towards the computational complexity estimates in Table 7.

In Table 8 we provide wall-clock timings of test-time prediction of the various methods using the Gemma 3 27B model. We report the summed time across the entire test-set with error bars across folds. For Semantic Entropy we also report the total number of LLM-based bidirectional entailment comparisons were performed, as the timing depends on the LLM being used. In general, this number is significantly smaller than the worst case (i.e. quadratic) since we use an efficient clustering approach, but the method is still orders of magnitude slower than all other methods.

## F  SOCIETAL IMPACTS & DISCUSSION

This work involves the development of the capabilities of large language models and there remain many open foundational challenges to the general safety of using these models Anwar et al., with potential broad societal risks. However, we believe that improving our ability to quantify the uncertainty of these models, in general, is an important step towards making them safer. Uncertainty quantification can play a key role in mitigating safety issues Rudner & Toner (2024), arising, for example, from out-of-domain generalisation or out-of-distribution data.

Large language models are compute intensive. The prevalent use of LLMs makes their carbon footprint an important consideration Patterson et al. (2021). Our method involves additional computation on top of the LLM in order to quantify uncertainty, specifically sampling multiple LLM responses to a prompt. However, we also point out that a specific contribution of this work is the reduction in the amount of extra computation required to quantify semantic uncertainty (in our case by 1000x over the best alternative method).

