# OpenReview forum: "SemanticDPP: Efficient Uncertainty Quantification in LLMs"
_ICLR.cc/2026/Conference — Submitted to ICLR 2026_

### Official Review · Reviewer_gKSj · 2025-10-24

**Soundness:** 1
**Presentation:** 3
**Contribution:** 3
**Rating:** 2
**Confidence:** 4

**Summary:**

The paper proposes a method for uncertainty estimation building on the Semantic Entropy work. The key contribution is to reduce inference-time costs by fitting a DDP (determinantal point process) to allow the entropy prediction based on just the model embeddings instead of the pairwise clustering in Semantic Entropy. The training process is an additional computational cost but could be (easily) amortized over inference calls.
They propose two different variants: SemanticDDP, which directly uses the learned DDP to estimate uncertainty from a set of model response embeddings and SemanticDDP-C, which uses the DDP to cluster samples responses into semantically coherent clusters.
The authors evaluate the method in six different benchmarks and with two different models: Gemma2 9B and Gemma3 27B.

**Strengths:**

- The paper identifies the bidirectional clustering step in Semantic Entropy as a speed bottleneck and alleviate it via replacing this step via their DDP method, while reporting even better results on the evaluated benchmarks. The usage of a DDP to this end is to the best of my knowledge novel.
- They discuss and raise an issue in benchmark datasets TriviaQA and NQ open, where answers at the benchmark creation time might have been true but has since changed.

**Weaknesses:**

See the questions. If the open questions and weaknesses raised there are appropriately addressed, I will raise my score.

nit: the caption formatting for many tables, where the caption has almost no vertical space to the table, is not very appealing.

nit: l.425 missing space between "SemanticDDPand"

**Questions:**

Q1. In Table 3, the scores in the columns do not add up to the average score -- e.g. for BioASQ `(0.82 + 0.56 + 0.6 + 0.63 + 0.53 + 0.86) / 6 = 0.667 != 0.658 (reported in paper)` or for TruthfulQA `(0.82 + 0.58 + 0.60 + 0.63 + 0.55 + 0.65) / 6 = 0.638 != 0.655 (reported in paper)`. How was the average score calculated?
  - this does change the interpretation of results, as the actual gap between fitting on BioASQ and TruthfulQA is now substantially larger, suggesting less transfer than was claimed in the paper

Q2. Why are the P(True) and Token Average Likelihood baselines not reported for Gemma3 37B in Table 3? Internal/implicit representations of uncertainty may scale in quality with model size and thus make the baselines more effective. Reporting these baselines is crucial to certain the method's effectiveness as model size scales.

Q3. What is the difference between Figure 2 and Table 1 & 2? It seems to be the exact same data minus Figure 2 not including the P(True) and Token Average Likelihoods baselines. Am I missing something? Also Table 1 and 2 are not referred to anywhere in the paper.

Q4. The computational overhead reported in your experiments for Semantic Entropy is really prohibitive, whereas the original paper (https://arxiv.org/pdf/2302.09664) claims the overhead is small in practice. Where does this discrepancy come from?

Q5. re: Table 4 -- what is the cumulative wall clock time of the actual LLM calls for the test split? This information better contextualizes how relevant the speedup in runtime of your reported methods is.

Q6. Which hyper parameters are used for the baselines, e.g. the "Linear probe" (Kossen et al., 2024)? Since the hyper parameters of the proposed method are tuned, are the baseline hyper-parameters also tuned (and how)?

---

### Official Review · Reviewer_s3Vm · 2025-10-31

**Soundness:** 3
**Presentation:** 2
**Contribution:** 2
**Rating:** 2
**Confidence:** 4

**Summary:**

The paper presents a new method for uncertainty quantification with LLMs. The idea is to train a function that maps the embedding of an input (e.g. in the last layer of the model) to a semantic space. The uncertainty of a new input is assessed by generating multiple outputs for a given input, mapping these to the semantic space with the learned function, and in the semantic space using a fitted kernel of a determinantal point process to quantify the variability/uncertainty of the model for the provided input.

**Strengths:**

I did not find any mistakes in the setup or experiments. The paper is relatively clearly written. The idea of training a model to predict uncertainties has a promise to improve efficiency. The method is tested with many datasets.

**Weaknesses:**

The novelty is somewhat limited. There are previous methods that train a model to predict the uncertainty from the model embedding, although the details differ. If there are differences in performance, it would be interesting to understand what exactly explains these. In general, I feel the conceptual difference to the closest methods would have warranted clearer discussion.

One claimed strength of the method is that it is faster, i.e., does not require the n^2 LLM queries to assess the semantic similarities between model outputs at test time. However, also the presented method requires n generated outputs. Generating the n outputs is in practice computationally much heavier than the pairwise comparisons (which are a simpler task that can be solved with a smaller model), and hence the main computational bottleneck of the previous methods remains (and this is not discussed clearly).

One weakness of the method is that the model that predicts the uncertainties from model embeddings has to be trained, and for the best performance this would ideally be done for each new dataset from different domains. This is not needed by the comparison methods (at least not all of them). I did not see many details of how much training is needed, and how the preformance of the method depends on the amount of training.

Limited evaluation: only two models are considered whereas recent works, e.g. Kossen et al., 2024; Nikitin et al., 2024, included 6 models (and basic/instruction-tuned variants). Also, recent relevant baselines are not included (e.g. Qiu and Miikkulainen 2024; Nikitin et al. 2024). Evaluation metrics are also more limited: earlier work used two metrics: AUROC for the classification accuracy of identifying incorrect answers and AUARC which measures accuracy of responding to questions after removing questions whose uncertainty exceeds given threshold (averaged over thresholds). It seems that here only the latter is reported?

**Questions:**

See weaknesses.

---

> ### Author Response · Authors · 2025-11-21
>
> We thank the reviewer for their time. We address each question below---please let us know if you have follow-up questions or concerns.
>
> 1) `The novelty is somewhat limited...`:
> * We highlight the differences between our methods, SemanticDPP and SemanticDPP-C, with the baselines we compare against in Section 5.1 Baselines.
>
> * Due to space limitations perhaps the discussion could be longer.   The most closely-related method is Eigenscore (Chen et al., 2024), where the authors compute their entropy based on the determinant of a kernel on internal model embeddings (also relying on a pre-trained LLM to produce the embeddings).  In the context of our setting, EigenScore is most similar to our No Projection SemanticDPP baseline, but with an important difference that we use our kernel given by Eqn 1 with the base DPP model to optimize the hyperparameter alpha and scaling parameters, i.e. length-scales, for each dimension—rather than compute a linear kernel.  A significant difference, and our belief for the performance difference, is that through formulating the approach under the DPP model, we use the marginal likelihood to learn a semantically meaningful subspace on which to fit our kernel (i.e. finding a non-linear mapping from the embeddings that isolates semantic meaning and ignores syntax, etc.).  It also provides the machinery to compute entropy as an actual normalized probability.
>
> 2) `One claimed strength of the method is that it is faster, i.e., does not require the n^2 LLM queries to assess the semantic similarities between model outputs at test time...`
>
> * Yes, generating N samples is admittedly computationally more expensive.  However, much of the cost can be saved due to k,v caching, and it is common practice to generate batched samples for a variety of methods (e.g. best-of-N, GRPO, etc.).  Thus we don’t believe this is prohibitive in practice across applications.  In addition, while the Semantic Entropy (Nature) paper states that one could use a smaller 1.5B DeBERTa model, they chose to use GPT3.5 because it performed the task better (from the paper: “We settle on using GPT-3.5 with the above prompt, as its entailment predictions agree well with human raters and lead to good confabulation detection performance.”).
>
> * We discuss in Section 2.1 (The Efficiency of Semantic Clustering using  Language Models) why the pairwise comparisons, even when done using a computationally efficient algorithm, is still practically very slow. Table 4 illustrates the computational bottleneck:
> Method  	    |       Runtime
> SemanticDPP        0.00186s
> SemanticDPP-C    29.6s
> Semantic Entropy 12914s x 16 ≈ 57.4hrs
> Linear Probe           4.8914s
> EigenScore              0.00148s
>
> * One argument in support of n generations from our empirical results demonstrates value in this approach: n-generation based uncertainty methods, like SemanticDPP, SemanticDPP-C, and Semantic Entropy, substantially outperform common single-response baselines like Linear Probe, P_true and Token Likelihoods on most tasks (Tables 1 & 2).
>
> 3) `One weakness of the method is that the model that predicts the uncertainties from model embeddings has to be trained...`
> * We agree with you that domain shift robustness is a useful feature of an uncertainty method. We carried out a preliminary investigation on the robustness of SemanticDPP in the presence of domain shift, and found promising initial results (please refer to Table 3: Task transfer performance (AUROC) of SemanticDPP using Gemma3 27B.). These results point us to an interesting area of follow-up work around exploring a paradigm for fitting a DPP that involves a mixture of different tasks to yield a more generalizable kernel.
> * Regarding training of the model, this is described by fitting a DPP in Section 4—we kindly ask that the reviewer refer to Section 4.1 for SemanticDPP and 4.2 for SemanticDPP-C.  We trained the model using early stopping on a validation set, which typically required on the order of ~100s steps or so of gradient descent.  By far the most expensive part of the model fitting process is generating embeddings for responses, which we precompute once.  Our claim is that once the model is trained this cost is amortized over the prediction of many examples.
>
> 4) `Limited evaluation: only two models are considered whereas recent works, e.g. Kossen et al., 2024; Nikitin et al., 2024, included 6 models (and basic/instruction-tuned variants)...`
> * Thank you for raising the point about performance metrics. We agree that computing AUARC would be an interesting comparison between the methods and will look into reevaluating our large baseline suite on this metric.

---

### Official Review · Reviewer_zF5f · 2025-10-31

**Soundness:** 3
**Presentation:** 2
**Contribution:** 2
**Rating:** 6
**Confidence:** 5

**Summary:**

This manuscript presents a new uncertainty quantification metric for LLMs. The main idea is to use determinantal point processes (DPPs) to learn a model that can quantify the semantic similarities of response embeddings. Its effectiveness is measured using several QA benchmarks under a selective prediction use case.

**Strengths:**

1.	Utilizing DPPs to quantify the uncertainty of LLMs is new and interesting, with solid mathematical foundations.
2.	The proposed mechanisms are complement to several existing approaches, having the potential to be combined with other methods to further improve the performance.
3.	The writing of the paper is transparent with regards to the limitations of the work.

**Weaknesses:**

1.	The returned uncertainty metric is prompt-wise, not response-wise. This may limit the use case, i.e., it cannot be used to decide which response is more trustworthy if we have multiple responses for the same prompt.
2.	The benefit in computation time is not clearly justified. The proposed approach still needs to sample multiple responses for each prompt. This cost may dominate other computation overheads.
3.	The empirical comparisons are only with prompt-wise uncertainty metrics in abstention setup.

**Questions:**

1.	It is stated that semantic entropy “is hampered by the prohibitive cost of requiring a second LM to compare pairs of sampled responses.” However, the NLI model used in semantic entropy is only 1.5B, which is significantly smaller than most other SOTA LLMs. The cost in running this NLI model should be negligible compared with the sampling cost of the original LLMs. Can you make this point clearer?
2.	When stating “where Y_i denotes the subsets corresponding to each prompt i”, do you mean each “Y_i” include multiple subsets of responses, and each subset contains responses with equivalent semantic meanings while different subsets represent distinct semantic meanings? Please make this description clearer.
3.	Regarding SemanticDPP-C, what is the use case for this clustering algorithm? Is it better to have more fine-grained or even continuous measurement of semantic similarities, like in Semantic Density [1]?
4.	Can you elaborate more on how the proposed approach is complement to Semantic Density [1] and how they can be combined together?
5.	Regarding the dataset issues, have you tried cleaning the dataset by removing the questions with out-dated answers? What are the resulting performances?
6. The results reported in Section 5.4 (Table 4) are confusing. What operations are included in this runtime for SemanticDPP? SemanticDPP still needs to sample multiple responses, right? Why this sampling time is not included in this total runtime?

[1] Xin Qiu, Risto Miikkulainen. Semantic density: Uncertainty quantification for large language models through confidence measurement in semantic space, Advances in Neural Information Processing Systems (NeurIPS), 2024

---

### Meta-Review · Area_Chair_tJp7 · 2026-01-09

**Summary:**

The paper proposes SemanticDPP, an uncertainty quantification method for LLM QA that replaces pairwise semantic-comparison, as in semantic entropy, with a DPP-based kernel over internal response embeddings. The direction is interesting and technically grounded. There are two main concerns of the paper. One is that the technical core has significant overlap with existing embedding-kernel log-determinant approaches—most notably EigenScore, and the performance improvement over semantic entropy is not overwhelming. Second and the most significant is that the claimed efficiency win over semantic entropy is not convincingly supported end-to-end. The paper is really on the boarderline of acceptance, and I recommend that the authors fix the second issue and resubmit.

**Reviewer Concerns:**

The submission remains borderline due to substantial outstanding concerns about (i) the strength of the efficiency claim, (ii) the completeness and clarity of the empirical evaluation, and (iii) the paper’s positioning relative to closely related embedding-based uncertainty methods.

**Reviewer Scores:**

Two reviewers may raise thier scores from 2 to 4.

---

### Decision · Program_Chairs · 2026-01-26

Reject